

# Toward autonomous surface-based infrared remote sensing of polar clouds: Retrievals of cloud microphysical properties

Penny M. Rowe[1,2], Christopher J. Cox[3,4], Steven Neshyba[5] and Von P. Walden[6]

[1]NorthWest Research Associates, Redmond, WA, USA
[2]Department of Physics, Universidad de Santiago de Chile, Santiago, Chile
[3]Cooperative Institute for Research in Environmental Sciences, University of Colorado, Boulder, CO, USA.
[4]NOAA Earth System Research Laboratory, Physical Sciences Division, Boulder, Colorado, CO, USA.
[5]Chemistry Department, University of Puget Sound, Tacoma, WA, USA.
[6]Department of Civil and Environmental Engineering, Washington State University, Pullman, WA, USA

*Correspondence to*: Penny M. Rowe (penny@nwra.com)

**Abstract**

Improvements to climate model results in polar regions require improved knowledge of cloud microphysical properties. Surface-based infrared radiance spectrometers have been used to retrieve cloud microphysical properties in polar regions, but measurements are sparse. Reductions in cost and power requirements to allow more widespread measurements could be aided by reducing instrument resolution. Here we explore the effect of errors and instrument resolution on cloud microphysical property retrievals from downwelling infrared radiances for resolutions of 0.1 to 8 cm$^{-1}$. Retrievals are tested on 331 radiance simulations characteristic of the Arctic, including mixed-phase, vertically inhomogeneous, and liquid-topped clouds and a variety of ice habits. Results indicate that measurement biases lead to biases in retrieved properties that are not represented by the retrieval error covariance matrix. Retrieval errors are high if mixed-phase is assumed throughout liquid-topped ice clouds. Errors due to assuming ice habit is spherical are progressively larger for solid columns, plates, and hollow bullet rosettes. Using retrieved cloud heights, particularly when errors are imposed, increases retrieval errors but decreases sensitivity to incorrect ice habits and vertical variation. Results indicate that retrieval accuracy is unaffected by resolution from 0.1 to 2 cm$^{-1}$, after which it decreases only slightly. At a resolution of 4 cm$^{-1}$, for typical errors expected in temperature (0.2 K) and water vapour (3%), and assuming radiation bias and noise of 0.2 mW/(m$^2$ sr cm$^{-1}$), using retrieved cloud heights, error estimates are 0.1 ± 0.6 for optical depth, 0.0 ± 0.3 for ice fraction, 0 ± 2 µm for effective radius of liquid, and 2 ± 2 µm for effective radius of ice. These results indicate that a moderately low resolution, portable, surface-based infrared spectrometer could provide microphysical properties to help constrain climate models.

## 1 Introduction

Knowledge of polar cloud properties is critical for understanding climate change in polar regions. Polar regions are among the most rapidly warming regions on Earth, with significant concurrent changes in cloud properties that influence the amount of warming (Wang and Key 2005) and indications that sensitivity to clouds may increase in a warming Arctic (Cox et al 2015). Clouds have a strong influence on the polar surface energy budget (Lawson and Gettelman 2014; van den Broeke et al 2017), influencing sea ice loss (Francis and Hunter 2006; Kay et al 2009; Wang et al 2011) and Greenland ice melt (van den Broeke et al 2017). Despite ongoing efforts to improve cloud processes in climate models, the Intergovernmental Panel on Climate Change (IPCC) finds that "clouds and aerosols continue to contribute the largest





uncertainty to estimates and interpretations of the Earth's changing energy budget," (Boucher et al, 2013). Improving the representation of cloud processes in climate models requires observational constraints, including ice and liquid water paths, particle size and thermodynamic phase (Komurcu et al 2014; Winker et al 2017). This is particularly true for the polar regions, where clouds and cloud processes are distinctly different from lower latitudes and present unique challenges for modeling cloud radiative effects (Hines et al 2004), and where surface-based measurements are challenging.

Although ground-based observations in the polar regions are sparse, measurements made during campaigns and at permanent field sites (e.g. Bromwich et al 2012; Cox et al 2014; Uttal et al., 2015, and references therein; Lachlan-Cope et al 2016; Silber et al 2018) and from satellites (e.g. L'Ecuyer and Jiang 2010) have made import contributions to our understanding of polar clouds. Infrared spectrometers are proven instruments for remote sensing that have been part of many of the surface and satellite-based measurements described above. Surface-based infrared spectrometers are most sensitive to the cloud base, providing an important complement to satellite-based measurements. In particular, Atmospheric Emitted Radiance Interferometer (AERI) instruments currently operate at Barrow (1998-current), Eureka (2006-current), and Summit (June 2010-current): three Arctic Intensive Observing Sites (Uttal et al., 2015). In the Antarctic, there have been only short-term surface-based infrared spectrometer measurements, including measurements made at Amundsen-Scott South Pole Station in 1992 (Mahesh et al 2001) and 2001 (Rowe et al 2008), at Dome C during Austral summer 2003 (Walden et al 2005) and 2012-2014 (Palchetti et al 2015), and McMurdo (as part of the Atmospheric Radiation measurement (ARM) West Antarctic Radiation Experiment, or AWARE; Silber et al 2018). These measurements are crucial, but represent only very sparse coverage of the polar regions.

Because infrared radiance measurements are passive, the energy requirements are considerably lower than for active instruments such as lidar. Thus there is the potential for portable, low-cost, autonomous infrared spectrometers that could be deployed to remote locations to make widespread infrared radiance measurements across the polar regions from which cloud microphysics could be retrieved. To demonstrate feasibility of such an instrument, the limitations of the retrieval given instrument operational constraints and availability of ancillary data must first be assessed.

In this paper, we explore the accuracy with which cloud properties could be retrieved from a portable infrared spectrometer, including optical depth, thermodynamic phase, and effective radius. This paper builds on similar work that explored the accuracy of cloud height retrievals (Rowe et al. 2016). One way to develop a robust, low-power portable spectrometer might be to reduce the instrument resolution. Here we quantify cloud-property retrieval accuracy resolution becomes coarser, from $0.1 \text{ cm}^{-1}$ to $8 \text{ cm}^{-1}$. Cloud properties are retrieved from simulated downwelling radiance spectra using the CLoud and Atmospheric Radiation Retrieval Algorithm (CLARRA), which uses an optimal inverse method in a Bayesian framework. Cloud property retrievals are performed for simulated polar clouds with varying atmospheric thermal and humidity structure, cloud optical depth in the geometric limit (hereafter, cloud optical depth or COD), thermodynamic phase (including mixed-phase and supercooled liquid), liquid effective radius, ice effective radius, ice crystal habit, and cloud vertical structure. We also examine the sensitivity of retrieved results on noise and bias imposed on the radiance, as well as on errors in specified input parameters, especially the atmospheric state and cloud height.



## 2 Simulated Radiances

To test the effect of instrument resolution on the ability to retrieve cloud properties from downwelling radiances, retrievals using CLARRA were performed on a set of simulations. Using simulations rather than actual measurements confers a variety of benefits: 1) the basic capability of the model, in the absence of error, can be determined, setting a benchmark for retrieval capability, 2) the effects of various sources of error (such as noise, bias, or uncertainty in the atmospheric state) can be determined and assessed independently, and 3) errors in the retrieved values are known and thus can be compared to assess the uncertainty prediction from the CLARRA model.

The set of simulated downwelling radiances used to test the retrieval is described in detail by Cox et al. (2016) and by Rowe et al (2016), and are thus described only briefly here. The simulations are based on observed Arctic atmospheric profiles and cloud properties meant to represent a typical Arctic year, based on statistics from field observations (Cox et al. 2016 and references therein; although designed for the Arctic, significant overlap is expected for typical Antarctic atmospheric states, except perhaps in winter in the interior, when the atmosphere is colder and drier). All clouds were modeled as plane-parallel, single-layer clouds.

A base set of 222 simplistic clouds was created; these clouds are vertically uniform and use spheres for ice crystal habit (as well as for liquid droplet shape). Cloud bases vary from 0 to 7 km, with about 70% of clouds within the lowest 2 km and 30% above; thickness varies from 0.1 to 1.6 km; and temperatures vary from 225 to 282 K. Mixed-phase clouds are modeled as externally mixed and span temperatures of 240 to 273 K. Cloud phase includes liquid-only (~1/6 of cases), ice-only (~1/6) and mixed-phase (~2/3). Statistics for cloud microphysical properties are summarized in Table 1 (statistics were generated for log-normal distributions of COD and effective radii). Thus the standard deviations were computed for the logarithms. For convenience, these were converted to positive and negative linear deviations in Table 1.

A second set of 92 cases was used to test the effects of cloud vertical inhomogeneity. To do this, 23 cases from the base set were selected for which the cloud spanned multiple layers of the atmospheric model; these are referred to as "diffuse." Simulations were then created for atmospheric and cloud conditions that were identical, including the total COD, except that clouds were modeled as dense (physically thinner), inhomogeneous (the cloud was optically thicker at the center and thinner at the upper and lower edges), or liquid-topped (liquid cloud was confined to the uppermost layer, while ice cloud was confined to the lower model layers).

A third set of 54 cases was used to test the effect of ice habit on the retrieval. For this purpose, 9 base cases were selected having an ice geometric optical depth greater than 0.5. Simulations were created for atmospheric and cloud conditions that were identical, except that single scattering properties from different cloud habits were used: hollow bullet rosettes, smooth plates, rough plates, smooth solid columns, and rough solid columns (Yang et al. 2005; 2013).

Infrared spectrometers operate at finite, rather than line-by-line, or perfect resolution. To determine the effect of reduced resolution on the cloud microphysical retrievals, resolutions of 0.1, 0.5, 1, 2, 4 and 8 cm$^{-1}$ were tested. To mimic finite-resolution measurements, the set of simulated spectra, which were created at perfect resolution, were convolved with a sinc function to obtain sets of spectra at each resolution.



Figure 1a shows a cloudy-sky radiance spectrum at 0.5 cm$^{-1}$ resolution, together with the clear-sky spectrum for the same atmospheric conditions. Additional examples at 0.5 cm$^{-1}$, as well at 4.0 cm$^{-1}$, are given in Fig. 2 of Rowe et al 2016.

### 3    CLoud and Atmospheric Radiation Retrieval Algorithm (CLARRA)

CLARRA retrieves cloud macrophysical properties (cloud height and temperature) and microphysical properties, (COD, ice fraction, effective radius of liquid droplets, and effective radius of ice droplets) from downwelling infrared radiances, given

knowledge of the atmospheric state.

As the first step in the retrieval, cloud heights are retrieved by CLARRA using the methods described in Rowe et al. 2016 (see also references therein). Alternatively, cloud heights can be input into CLARRA (e.g from other instrumentation, such as lidar, or from reanalysis models). Next, CLARRA computes the cloud emissivity, neglecting scattering, and retrieves

from it first-guess estimates of cloud microphysical properties. Finally, the microphysical retrieval is performed using an iterative optimal nonlinear inverse method (Rodgers et al. 2000).

In preparation for running CLARRA, model atmosphere layer boundaries must be chosen and the atmospheric profiles must be constructed (based on model and measured data for the location and time of the downwelling radiance spectrum). For this

work, the same atmospheric profiles used to create the simulated radiances are used (although errors are sometimes added). In addition to uncertainty estimates for the dowelling radiance, the microphysical retrieval requires *a priori* values for the microphysical properties and their covariance matrix. These can be taken from a climatology or can be determined from the fast retrieval. In this work, the statistics of the cloud properties used to create the simulated radiances are used.

Finally, the observed spectrum and associated covariance matrix are needed (here, the simulated radiances with known errors are used). After these preparations, CLARRA is run as follows.

1.  Compute gaseous layer optical depths at monochromatic resolution.
2.  Using the above and the temperature profile, calculate terms related to emission and transmission by gases at the effective instrument resolution.
3.  Retrieve cloud height (see Rowe et al 2016), or alternatively, input the cloud height from another source.
4.  Perform the fast retrieval that neglects scattering, to get first-guess microphysical properties.
5.  Retrieve the microphysical properties for the next iteration using the current retrieval estimates, the *a priori* and covariance matrix for the microphysical properties, and the observed spectrum and its covariance matrix.
6.  Repeat step 5 until the result converges or a maximum number of iterations is reached.

For step 1, gaseous layer optical depths are computed at monochromatic resolution using the Line-By-Line Radiative Transfer Model (LBLRTM; Clough et al 2005). Remaining steps are described below.

### 3.1    Resolution and Model Errors

The emission and transmission terms required for step 2 must be at an effective resolution consistent with the resolution of

the observed spectra; i.e. the instrument resolution. Calculating each required term at monochromatic resolution and then reducing the resolution would be extremely time intensive. Instead, it is desirable to use effective-resolution quantities that





can be computed quickly with minor loss in accuracy. Because COD varies slowly with wavenumber, in exploring the best methods for resolution-matching in CLARRA, we focus on gaseous optical depths, which vary rapidly with wavenumber. This section describes the process of computing effective-resolution terms required for the cloud height and microphysical
property retrievals.

To solve the radiative transfer equation in LBLRTM and the DIScrete Ordinates Radiative Transfer code (DISORT; Stamnes et al 1988) the atmosphere is divided into model atmospheric layers and the approximation is made that the Planck function varies linearly with optical depth through the layer (Wiscomb et al 1976, Clough et al 1992). In the absence of
scattering, the downwelling radiance from a layer at a given wavenumber is approximated as

$$\Delta \tilde{R}_L = \int_{\tilde{\tau}_{L-1}}^{\tilde{\tau}_L} \tilde{B}(\tilde{\tau}) e^{-\tilde{\tau}\sec\theta} d\tilde{\tau} \tag{1}$$

where the tildes indicate monochromatic, or perfect, resolution (all quantities with tildes depend on wavenumber), $\tau$ is
defined as the vertical optical depth from the surface up to some height (e.g. within layer $L$), $\tau_{L-1}$ is from the surface to the layer bottom, and $\tau_L$ is from the surface to the layer top. $\tilde{B}$ is the Planck function and $\theta$ is the viewing angle from zenith. Note that the formulation here differs from that of Clough et al (1992); here, $R_L$, $\tau_L$, and the transmittance, $t_L$ (defined below) are defined from the bottom of the model atmosphere (e.g. from Earth's surface), to the top of layer $L$. Quantities that are for layer bottom to top only are indicated with a delta. Using these conventions means that Eq. (1) represents the
radiance from layer $L$ that is transmitted by the atmosphere below to the surface. The viewing angle is included explicitly here so that $\tau$ refers to the vertical optical depth.

The linear-in-optical depth approximation for $B$ allows the integral to be solved, yielding

$$\Delta \tilde{R}_L = -\tilde{B}_L \tilde{t}_L + \tilde{B}_{L-1} \tilde{t}_{L-1} - \Delta \tilde{B}_L \left[\frac{\Delta \tilde{t}_L}{\Delta \tilde{\tau}_L}\right] \tag{2}$$

where $\tilde{B}_{L-1}$ and $\tilde{B}_L$ are the Planck functions of the temperature at the lower and upper boundary of layer $L$, the deltas indicate the change across the layer, and the surface-to-layer top transmittance is

$$\tilde{t}_L = \exp(-\tilde{\tau}_L \sec\theta). \tag{3}$$

Thus $R_L$ is the radiance from the layer that makes it to the surface.

Since real-world instruments operate at finite resolution, the recorded spectrum is the layer radiance convolved with the
instrument lineshape $S$,

$$\Delta R_L(v) = \int_{-\infty}^{\infty} \Delta \tilde{R}_L(\tilde{v}) S(v, \tilde{v}) d\tilde{v}, \tag{4}$$



where the dependence on wavenumber has been included explicitly. In practice the integral need only be performed over the small wavenumber region characterized by the width of $S$ (typically a sinc function). The total downwelling radiance up to each potential cloud layer ($c$) is calculated as the sum of layer radiances,

$$R_c = \sum_{L=1}^{c} \Delta R_L. \tag{5}$$

The cloud-height retrieval (Rowe et al 2016) requires calculations of the gaseous radiance from the surface up to each possible cloud layer ($R_c$), the clear-sky radiance ($R_{clr}$), and the Planck function at the temperature of each possible cloud times the transmittance of the atmosphere below the cloud ($B_c t_c$). To calculate $R_c$ and $R_{clr}$, first gaseous layer optical depths $\Delta \tilde{\tau}_L$ are computed using LBLRTM (step 1 above). Next, for each layer, these are summed over all the layers below to get $\tilde{\tau}_L$. Equations (2) - (5) are then used to calculate $\tilde{t}_L$ and $R_N$. This is repeated for each model layer $c$ that could contain cloud. $R_{clr}$ is computed in the same way but with the top layer of the model atmosphere used in place of $c$. (Note that $\tilde{R}_L$ is calculated slightly differently than in Eq. (2) in that a Padé approximation is used, following Clough et al. 1992; the two methods give very similar results). For a cloud in layer $c$, the product ($B_c t_c$) at the effective instrument resolution is

$$(B_c t_c)(v) = \int_{-\infty}^{\infty} B_c(\tilde{v}) \tilde{t}_c(\tilde{v}) S(v, \tilde{v}) d\tilde{v}. \tag{6}$$

Figure 1b shows model errors for a typical clear-sky radiance used in the cloud height retrievals ($R_{macro, clear}$); the error shown is the difference between the clear-sky downwelling radiance calculated as described above and the monochromatic radiance from LBLRTM convolved with the instrument lineshape. Model errors for clear skies are typically very low, as in Figure 1b.

Retrieval of microphysical cloud properties requires effective-resolution layer optical depths, $\Delta\tau_L$, as input into the DISORT radiative transfer code. One method to create the set of $\Delta\tau_L$ might be to reduce the resolution of the layer optical depths. However, examination of Eqs. (2) – (4) suggests that a more accurate method would be in terms of $t_L$. Inserting Eq. (2) into Eq. (4) and breaking up the integral gives

$$\Delta R_L(v) = -\int_{-\infty}^{\infty} \tilde{B}_L \tilde{t}_L S(v,\tilde{v}) d\tilde{v} + \int_{-\infty}^{\infty} \tilde{B}_{L-1} \tilde{t}_{L-1} S(v,\tilde{v}) d\tilde{v} - \int_{-\infty}^{\infty} \Delta\tilde{B}_L \left[\frac{\Delta\tilde{t}_L}{\Delta\tilde{\tau}_L}\right] S(v,\tilde{v}) d\tilde{v}. \tag{7}$$

From Eq. (6), we see that the first two terms in Eq. (7) are $-(B_L t_L)$ and $(B_{L-1} t_{L-1})$. This suggests that we define an effective layer optical depth in terms of these quantities. First, we define an effective transmittance,

$$t_L \equiv (B_L t_L)(v)/B(T_{ave,L}, v). \tag{8}$$

Because the transmittance is defined from the surface, to get the optical depth of a layer, the optical depths of all layers below must be subtracted. The optical depths are therefore computed layer by layer, starting with the surface. For the first layer,

$$\Delta\tau_1 \equiv -\log(t_1). \tag{9}$$



For subsequent layers,

$$\Delta\tau_L \equiv -\log(t_L) - \sum_{x=1}^{L-1}\Delta\tau_x. \tag{10}$$

Due to ringing, effective-resolution transmittances can be greater than 1 or less than 0, resulting in optical depths outside physical bounds. To minimize ringing, radiances and transmittances were averaged over small spectral regions between strong emission lines, or microwindows, before applying Eqs. (9) and (10). Following this, transmittances below $10^{-40}$ and above 1 were modified such that $10^{-40} <= t_L <=1$. Finally, since DISORT is run at single precision, serious errors can result for very small input optical depths, thus $\Delta\tau_L$ was increased as needed such that $\Delta\tau_L >= 10^{-5}$. The $\Delta\tau_L$ were then used as inputs to DISORT to perform the radiative transfer.

Figure 1b also shows box and whiskers plots of model errors for cloudy-sky radiances at 0.5 cm$^{-1}$ resolution, averaged within microwindows (see Supplemental) used in the cloud microphysical property retrievals. The errors were calculated as differences between downwelling radiance calculated using the effective-resolution layer optical depths as described above and monochromatic radiances convolved with the instrument lineshape (the radiance simulations described in Sect. 2. For all microwindows, median model errors are within +/- 0.05 RU, 3/4 of model errors are within +/- 0.10 RU, and all model errors are within +/- 0.3 RU. Overall, errors are similar at finer spectral resolutions (0.1 cm$^{-1}$) but get larger as resolution gets coarser with a range of -0.2 to 0.8 RU at 8 cm$^{-1}$ resolution. (Figures similar to Fig. 1 are shown at other resolutions in the Supplemental).

### 3.2 Fast preliminary retrieval: The first-guess

The fast retrieval neglects scattering and uses the absorption approximation to allow fast computation of a first-guess set of microphysical properties. The cloud reflectivity is ignored so that the emissivity is assumed to be one minus the cloud transmittance. The emissivity is approximated as in Rowe et al (2016)

$$\varepsilon = \frac{R_{obs} - R_{clr}}{B_c t_c + R_c - R_{clr}}, \tag{11}$$

where $R_{obs}$ is the observed radiance (here, the simulated radiances are used) at the instrument resolution. Calculation of other terms is described above. The cloud absorption optical depth can then be calculated from quantities that are measured or can be calculated independently of the cloud microphysical properties:

$$\tau_{a,obs} = -\ln\left(1 - \frac{R_{obs} - R_{clr}}{B_c t_c + R_c - R_{clr}}\right). \tag{12}$$

The value of $\tau_a$ can also be calculated from the state variables: COD ($\tau_g$), ice fraction ($f_{ice}$), effective radius of liquid ($r_{liq}$), and effective radius of ice ($r_{ice}$),

$$\tau_a = \tau_g/2 \; [\; [1\text{-}f_{ice}] \; Q_{a,liq}(r_{liq}) + f_{ice} \; Q_{a,ice}(r_{ice}) \;\;]. \tag{13}$$



$Q_{a,liq}$ and $Q_{a,ice}$ are the absorption efficiencies of liquid and ice, determined from the extinction efficiencies $Q_e$ and the single
scatter albedos $\omega_0$. For ice

$$Q_{a,ice} = Q_{e\_ice}(r_{ice})\,[1-\omega_{0,ice}], \tag{14}$$

where $Q_{e,ice}$ and $\omega_{0,ice}$ are determined for averages over a log-normal distribution of particle radii corresponding to the
effective radius $r_{ice}$. For the fast preliminary retrieval, spheres were assumed for ice and single scattering parameters for
each particle radius were calculated from Mie theory using the index of refraction of Warren et al (2008), based on a
temperature of 266 K. Both spherical and realistic ice habits were used in the full retrieval. Scattering properties for hollow
bullet rosettes and smooth and rough solid columns and plates were provided by Yang et al (2013). For liquid, single-
scattering parameters determined from temperature-dependent indices of refraction at temperatures of 240, 253, 263, and
273 K were used (Rowe et al 2013; Zasetsky et al 2005; Wagner et al 2005). Letting $T_1$ be the temperature from this list that
is closest to but lower than the cloud temperature, and $T_2$ be the temperature closest to but higher than the cloud temperature,
$Q_{a,liq}$ is given as the weighted sum

$$Q_{a,liq} = w_1\,Q_{e\_liq}(r_{liq},T_1)\,[1-\omega_{0,liq}\,(r_{liq},T_1)] + w_2\,Q_{e\_liq}(r_{liq},T_2)\,[1-\omega_{0,liq}\,(r_{liq},T_1)], \tag{15}$$

where $w_1 = (T_2 - T_c)/\,(T_2 - T_1)$ and $w_2 = (T_c - T_1)/\,(T_2 - T_1)$.

The values of $Q_{e,liq}$ and $Q_{e,ice}$, $\omega_{0,liq}$ , and $\omega_{0,ice}$, are pre-computed for the full range of possible $r_{liq}$ and $r_{ice}$. The COD ($\tau_g$) is
retrieved by inverse retrieval (using Eqs. (16) and (17) given below, but with **R** replaced with $\tau_{a,\,obs}$, **F(x)** replaced with Eq.
(13) and $\gamma = 0$). Next, $\tau_{a,\,obs}$ is calculated for the retrieved $\tau_g$ and for a variety of values of $f_{ice}$, (0.2, 0.4, 0.6, 0.8), $r_{liq}$
(integers between 5 and 30) and $r_{ice}$ (even numbers between 10 and 50). Calculating $\tau_{a,\,obs}$ for all combinations of these
values is computationally fast compared to other aspects of CLARRA. Finally, the values of $f_{ice}$, $r_{liq}$, and $r_{ice}$ are selected that
correspond to the minimum absolute difference between $\tau_{a,obs}$ and $\tau_a$.

The fast retrieval serves as a first guess for the inverse retrieval including scattering. The use of a first guess helps stabilize
the retrieval and reduces computation time.

### 3.3    Optimal Nonlinear Inverse Method

The optimal nonlinear inverse method uses radiances from 400 to 600 cm$^{-1}$ (allowing thermodynamic phase determination;
Rathke et al. 2002) and from 750 to 1300 cm$^{-1}$, which is sensitive to phase, COD and effective radius. The inverse method is
similar to the method of Turner (2005), which has been used to retrieve cloud properties from AERI instruments in the
Arctic (Turner 2005; Cox et al 2014). Similar inverse methods have also been used from satellite instruments (L'Ecuyer et
al 2006; Wang et al 2006). The inversion equation used here is the iterative Levenberg-Marquardt method
{*Rodgers:2000tw and references therein},

$$\mathbf{x}_{i+1} = \mathbf{x}_i + \left\{ [1+\gamma_i]\mathbf{S}_a^{-1} + \mathbf{K}_i^T \mathbf{S}_e^{-1} \mathbf{K}_i \right\}^{-1} \left\{ \mathbf{K}_i^T \mathbf{S}_e^{-1} \left[ \mathbf{R} - \mathbf{F}(x_i) \right] - \mathbf{S}_a^{-1} \left[ \mathbf{x}_i - \mathbf{x}_a \right] \right\}, \tag{16}$$



where **x** is the state vector, with *a priori* $\boldsymbol{x_a}$ and covariance matrix is $\boldsymbol{S_a}$. $\boldsymbol{R}$ is the observed radiance spectrum, with covariance matrix $\boldsymbol{S_e}$. The forward model **F** is the discrete ordinates radiative transfer model (DISORT; Stamnes et al 1988). The subscript $i$ indicates the iteration number, and $\gamma_i$ is a weighting parameter, discussed below. The kernels (**K**) are the Jacobians, computed numerically by perturbing each state variable in turn. For example, for the first state variable, $x_1$, **K** is computed as

$$K(x_1) = \frac{R(x_1, x_2, \ldots) - R'(x_1', x_2, \ldots)}{x_1 - x_1'} \quad . \tag{17}$$

where primes indicate that a perturbation was applied. After computing **K** for each $x_i$, these are combined into a matrix with rows $K(x_1)$, $K(x_2)$, etc.

The Levenberg-Marquardt formulation given in Eq. (16) is a hybrid of the Gauss-Newton formation and the method of steepest descent. The parameter $\gamma$ weights the two methods: a value of $\gamma = 0$ defaults to Gauss-Newton. As $\gamma$ increases, the retrieval becomes more heavily weighted towards steepest descent and convergence slows. Choosing $\gamma_i$ for each iteration is difficult, as a large value of $\gamma$ will slow the retrieval. Here we use a conservative approach, starting with $\gamma_0 = 0$. If the first iteration causes error to increase by more than a specified threshold, the retrieval is repeated with $\gamma_1 = 1$. For following iterations, if error increases more than the threshold, $\gamma_{i+1}$ is increased to 10 $\gamma_i$, while if it doesn't, $\gamma_{i+1}$ is decreased to 0.1 $\gamma_i$. Iterations are repeated until $\gamma_i < 0.01$ and convergence is achieved, or until the maximum allowed number of iterations is reached. The error threshold was set by trial and error to be an increase in magnitude of the rms error between measurement and forward model by more than 1 RU or by more than double the current rms error.

Error in the retrieved state variable is given by the covariance matrix

$$\mathbf{S} = (\mathbf{K}^T \mathbf{S}_e^{-1} \mathbf{K} + \mathbf{S}_a^{-1})^{-1}, \tag{18}$$

and convergence is tested using

$$d_i^2 = (\mathbf{x}_i - \mathbf{x}_{i+1})^T \mathbf{S}^{-1} (\mathbf{x}_i - \mathbf{x}_{i+1}) \ll n, \tag{19}$$

(Rodgers 2000), where $n$ is the length of **x**.

In this work, the "observation" **R** is derived from the simulated spectra described in Sect. 2 by averaging the simulated radiances in the microwindows. Selected microwindows (see Supplemental) are similar to those in Turner (2005) and Garrett and Zhao (2013). Microwindows typically span at least 4 cm$^{-1}$, so that at least one simulated radiance falls in each microwindow, except at a resolution of 8 cm$^{-1}$. Thus for retrievals at 8 cm$^{-1}$, microwindows were expanded to be at least 8 cm$^{-1}$. The state vector **x** is composed of COD, ice fraction, log of the effective radius of liquid, and log of the effective radius of ice, so that $n = 4$. For the *a priori* ($\mathbf{x}_a$), means of the true values of **x** are used (Table 1). The covariance matrix $\mathbf{S}_a$ is





assumed to be diagonal, with diagonal elements based on a standard deviation of about one half the range of values; this is
used rather than using the standard deviations given in Table 1 in order to weight the retrieval heavily toward the
measurement rather than the *a priori*. The error covariance matrix for radiance ($\mathbf{S}_e$) is assumed to be diagonal with elements
based on the model errors described in the previous section and the measured and simulated radiance errors due to any
imposed errors, added in quadrature. The first guess values ($i = 0$) are determined from the fast microphysical property
retrieval. The maximum number of allowed iterations was set to 20 and the tolerance for convergence was set to $d^2 < 1$. For
convenience, the result of the forward model acting on the retrieved state vector is termed the retrieved radiance.

The forward model (F) is calculated by running DISORT with the state variables and the effective-resolution optical depths
described previously as input. Other inputs to DISORT include the solar contribution, surface albedo, temperature profile,
and the Legendre moments, single-scatter albedo, and COD, which depend on the state variables and cloud height.

## 4   Imposed Errors

In order to determine the impact of sources of error on the microphysical retrievals, various errors were imposed on
"observed" radiances, including Gaussian noise ($\pm$ 0.2 RU) and bias (0.2 RU). In remote locations, reanalysis datasets may
be used for specification of the atmospheric state. Wesslen et al. (2014) characterized temperature errors in the European
Centre for Medium-range Forecasting (ECMWF) Interim (ERA-Interim; Dee et al., 2011) as varying from -0.5 K to 1 K.
Rowe et al (2016) found such error to have a roughly equivalent effect on radiative transfer calculations as a positive
temperature bias of 0.2 K (Rowe et al 2016). Wesslen et al. (2014) characterized water vapor errors to be 2% to 10%, with
lower biases in the first 3 km and higher biases above. Because PWV decreases rapidly with height, this was found to be
roughly equivalent to a PWV bias of 3% (Rowe et al 2016). Based on this work, imposed errors also included errors in the
atmospheric temperature ($\pm$ 0.2 K), and the precipitable water vapor, or PWV ($\pm$ 3%). In addition, higher errors in PWV
were tested ($\pm$ 10%). Microphysical properties were retrieved with these errors each imposed in isolation, using both true
cloud heights and cloud heights retrieved with $CO_2$ slicing as described in Rowe et al (2016).

In addition to errors imposed in isolation, various combinations of errors were used:

a) noise of 0.2 RU, radiation bias of 0.15 RU, and PWV bias of -3%, with true cloud heights;

b) identical errors but with opposite signs on the PWV and radiation bias errors;

c) identical errors as in (a) except with retrieved cloud heights;

d) identical errors as in (b) except with retrieved cloud heights;

e) noise of 0.1 RU, temperature bias of 0.1 K, with true cloud heights;

f) noise of 0.1 RU, temperature bias of 0.1 K, with retrieved cloud heights;

g) noise of 0.2 RU, radiation bias of 0.2 RU, temperature bias of 0.2 K, and PWV bias of 3%, using true cloud heights;
and

h) noise of 0.2 RU, radiation bias of 0.2 RU, temperature bias of 0.2 K, and PWV bias of 3%, using retrieved cloud
heights.

Errors in retrieved cloud heights with (and without) the same imposed errors are given in Rowe et al (2016). Two cloud-
height retrieval algorithms were used in Rowe et al (2016); here we focus exclusively on results for the $CO_2$ slicing method.
Cloud-height errors were found to differ markedly for clouds with bases below 2 km ("low clouds") and above 2 km ("high





clouds"). For example, in the absence of error, at a resolution of 0.5 cm$^{-1}$, retrieved cloud heights were found to be biased by

0.2 km for low clouds and by -0.1 km for high clouds. For low clouds, positive radiation bias (0.2 RU) or negative temperature bias (-0.2 K) cause negative biases in cloud height that effectively cancel out the 0.16 km bias. In contrast, biases with reverse signs roughly double cloud-height retrieval bias. Noise and biases in PWV have little effect on biases in retrieved cloud height. Overall, for low clouds with the imposed errors described here, mean and standard deviations of cloud-height error are 0 to 0.3 km ± 0.3 to 0.4 km. For high clouds, errors in retrieved cloud height are much greater. Mean

and standard deviations of cloud-height error are -0.8 to 0.5 km ± 0.5 to 1.7 km for PWV biases and -1.4 to 1 km ± 0.5 to 1.8 km for radiation biases, temperature biases, or combined errors a and b.

## 5    Results and Discussion

### 5.1    Retrieval overview

Figure 2 shows error contours for an ice-only cloud with a COD of 0.89 and effective radius of 22 µm. Error contours are the root-mean-square differences between the spectra and the forward model applied to the retrieved variables. Superimposed on the error contours is the retrieval trajectory with iteration, starting from the first-guess value (red dot on right in each panel). As the figure shows, the retrieval converged in 4 iterations to the minimum. Furthermore, the retrieval correctly converged to an ice-only cloud, although the mean cloud temperature of ~256 K falls within the range of

temperatures where mixed-phase clouds may occur, and the first-guess ice fraction was 0.8.

Retrievals using the base set of simulations indicate that the kernels are typically sufficiently linear to converge on the solution, except for large COD and effective radii. The radiative transfer loses sensitivity to COD between about 6 and 10 (see Sect. 5.2 below) and loses sensitivity to effective radius above about 50 µm (see Garrett and Zhao 2013 and

Supplemental). In addition, when values approach these limits, the retrieval was found to sometimes move away from the solution. In addition to setting upper bounds for the COD and effective radius, the kernels were typically calculated for a step in the direction of smaller COD and effective radius; that is, in the direction where sensitivity is larger.

Nearly all retrievals converged to within the specified tolerance in $d^2$, with only 0 to 2 cases failing to converge for any set

of imposed errors. Overall, convergence was achieved in a mean of 4 iterations (median of 3). For retrievals performed for base cases for varying levels of imposed error, retrievals generally converged within 20 iterations (at most 2 cases failed to converge for any set of imposed errors).

The effect of using the fast retrieval result as a first guess was also examined. The fast retrieval was found to provide a first-guess value that is generally closer to the solution than the *a priori*. Compared to using the *a priori*, using the fast retrieval

result as the first guess reduces the rms error in the first guess from 300% to 6% for $\tau_g$, from 0.4 to 0.2 for $f_{ice}$, from 4.4 to 3.7 µm for $r_{liq}$ and from 16 to 11 for $r_{ice}$. (The first-guess values for $\tau_g$ and $r_{ice}$ were not allowed to exceed 5 and 50 µm, respectively). Use of the fast retrieval lowered retrieval errors slightly and modestly increased the number of cases that converged (typically an increase of no more than 1 case). In rare cases, using the fast retrieval can avoid spurious results with large errors due to incorrectly converging (this occurred one or fewer times for each set of retrievals). Overall, the

greatest improvement from using the fast preliminary retrieval is computational speed. On average, one fewer iteration was needed when the fast retrieval was used.



### 5.2   Retrieval Errors

To determine the retrieval capability, errors in retrieved values are examined in the absence of any imposed errors, where
only model errors are present. Table 2 shows errors in retrieved cloud microphysical properties ($\tau_g$, $f_{ice}$, $r_{liq}$, and $r_{ice}$) for the
base set of spectra, for spectral resolutions of 0.1, 0.5, and 4 cm$^{-1}$. Results were omitted where the root-mean-square
difference between observed and retrieved spectra was greater than 1 RU; this resulted in omitting a single point for 4 cm$^{-1}$
resolution. Retrieval errors are shown for different ranges of $\tau_g$. For thin clouds ($\tau_g < 0.4$), the signal-to-noise is low. For
thick clouds ($\tau_g > 5$), the spectrum begins to approach saturation, and sensitivity to the cloud's microphysical properties
diminishes. Both can result in large errors in $f_{ice}$, $r_{liq}$, and $r_{ice}$. (Error in $\tau_g$, by contrast, increases with increasing $\tau_g$ and is
smallest for the thinnest clouds). Based on these considerations, the ideal range for $\tau_g$ was identified as $0.4 < \tau_g < 5$. Unless
otherwise specified, results will be presented for this range. Retrieval errors for $0.4 < \tau_g < 5$ are overall quite low, with
magnitudes of errors in $\tau_g$ below 0.02, in $f_{ice}$ below 0.12, in $r_{liq}$ below 1.2, and in $r_{ice}$ below 5. The table shows a slight
increase in error with coarsening resolution, which is most pronounced for $f_{ice}$ and $r_{liq}$.

Errors in retrieved microphysical cloud properties for different imposed errors are given in Table 3 for a spectral resolution
of 0.5 cm$^{-1}$ and $\tau_g$ between 0.4 and 5. Magnitudes of imposed errors are given in the first column except for cases of
combined errors, which are detailed in Sect. 4. Subsequent columns give the means and standard deviations of the errors.
Cases are again omitted when the root-mean-square difference between the observed and retrieved radiance is greater than 1
RU; this resulted in 12, 9, and 2 points being removed for PWV biases of 10% and -10% and set d of combined errors
(respectively). When true cloud heights are used, errors in $\tau_g$ are within $\pm$ 0.2 for large PWV bias ($\pm$ 10%) or combined
errors, and within $\pm$ 0.04 for all other imposed errors (one standard deviation). When cloud heights are retrieved (columns
labelled CO$_2$ slicing and combined errors c and d), biases imposed on radiation, temperature, and PWV cause biases in
cloud temperature. The biases in cloud temperature, in turn, increase mean biases (to 0.1) and standard deviations (to 0.3 to
05) in $\tau_g$. This is because cloud emissivity depends fairly linearly on $\tau_g$, so that spectrally flat errors have the largest effect
on $\tau_g$; biases in cloud temperature cause errors that are spectrally fairly flat. In comparison to errors in $\tau_g$, errors in $f_{ice}$, $r_{liq}$,
and $r_{ice}$ don't vary as much with type of imposed error, but rather vary with magnitude of error. For large biases, it becomes
difficult to distinguish liquid and ice, with errors in $f_{ice}$ as large as 0.4 (absolute value of mean $\pm$ standard deviation), and
corresponding errors in $r_{liq}$ and $r_{ice}$ as large as 6 and 15 $\mu$m. Errors in $r_{ice}$ are 2 to 3 times as large as errors in $r_{liq}$. Given that
errors in habit are ignored for the base case, and given the ice fractions can be low in real clouds, this suggests that
uncertainties in retrieved $r_{ice}$ for real clouds will be large. Taken together this error analysis demonstrates the importance of
minimizing bias errors as much as possible.

Errors in retrieved state variables as a function of resolution, from 0.1 to 8 cm$^{-1}$ are shown in Fig. 3 for the base set. Mean
errors and standard deviations are shown for base cases with no imposed error and for combination c of imposed errors:
relative humidity errors of -3%, noise of 0.2 RU, radiation bias of 0.15 RU and CHR errors in cloud height. In the absence
of imposed errors, retrieval errors increase slightly overall with coarsening resolution. For imposed errors, no trend is seen
in retrieval errors for resolutions of $0 - 2$ cm$^{-1}$, after which standard deviations increase about 10 to 60%, with little change
in mean errors except in $\tau_g$, which roughly triples. Thus an instrument resolution of 2 cm$^{-1}$ seems to be a good compromise
between reducing resolution avoiding increased errors.



For comparison to Rowe et al (2016), retrieval errors were estimated for a surface-based infrared radiance spectrometer at 4 cm$^{-1}$ resolution with a radiation bias of 0.2 RU and noise of 0.2 RU. Since reanalysis data would likely be used to characterize the atmospheric state, error estimates discussed previously were used for errors in temperature (bias of 0.2 K) and PWV (bias of 3%). (This is error combination h, described in Sect. 4). Furthermore, it was assumed that no independent

data for cloud heights would be available, so cloud height was retrieved using $CO_2$ slicing. For the same cases and imposed errors, Rowe et al (2016) reported errors that increase from approximately 0.5 to 1.5 km as cloud-base height increases from within 1 km to 4 km. Errors in retrieved microphysical cloud properties for these imposed errors were found to be 0.1 ± 0.6 for optical depth, 0.0 ± 0.3 for ice fraction, 0 ± 2 μm for effective radius of liquid, and 2 ± 2 μm for effective radius of ice.

### 5.3    Retrieval error covariance matrix

Discussion of errors so far has focused on actual retrieval errors, can be calculated because simulated data was used as the observation set. For real measurements, error analysis relies on the covariance matrix $S$, which in turn depends on the kernels and covariance $S_e$ (Eq. (18)). Recall that $S_e$ depends on measurement and model errors. Here we determine how well the covariance matrix $S$ represents errors, and if the model error estimate is sufficient. For unbiased, normally distributed errors, the diagonals of $S$ should correspond to the 68% confidence interval. We can test this by determining how many of

the squared errors fall within the variance given by $S$. This is complicated by the fact that $S$ is not constant but depends on $x$ (because the kernels depend on $x$). Thus for each retrieved $x$, the absolute error was divided by the square root of the appropriate diagonal element of the corresponding $S$. For Gaussian errors, this ratio should be <=1 for 68% of retrievals (and <=2 for 95% of retrievals, etc). In the absence of imposed error, only 52 to 63% of retrievals had a ratio within 1 (for $S_e$ based on model errors calculated as shown in Fig. 1). Model error was therefore increased 0.02 RU to give values of 61 to

70%. It is important to note that the error distributions decrease more slowly than Gaussians, with only 81 to 87% of errors (rather than 94%) falling within the second standard deviation indicated by $S$. In retrievals from real observations, additional sources of model error should be included, such as model errors in DISORT and LBLRTM.

For imposed noise of 0.2 RU, only 52 to 58% of retrievals have a ratio within 1, indicating that model errors are amplified in

the presence of errors. This is likely because away from the correct solution, the estimate of $S$ is incorrect. Increasing the contribution of noise to $S_e$ by 25 to 30% accounted for this, resulting in values of 62 – 77%.

The inverse retrieval is based on an assumption of unbiased, normally distributed errors, so $S$ provides a poor indication of retrieval errors due to bias errors in radiance, temperature, PWV, or cloud height. For biases in radiance, PWV, and for

errors in cloud height $S$ is particularly non-representative for COD, for which only 11 to 25% of cases fall within one standard deviation for $S$ (for other properties the range is 36 to 78%). Biases in temperature affect $S$ similarly for all properties (range of 48 to 66%). This underscores the importance of removing bias errors from measurements whenever possible to ensure that $S$ provides the best possible representation of errors.

### 5.4    Cloud vertical inhomogeneity and Ice habit

In addition to understanding how errors affect the retrievals for the base set of idealized clouds, discussed previously, it is import to understand what effects variability in clouds will have on the retrieval. For this purpose we use the set of 92 cases used to test the effects of cloud inhomogeneity, described in Sect. 2, which are comprised of 23 cases each of dense, diffuse, inhomogeneous (thinner at the bounds) and liquid-topped clouds.



Errors in retrieved microphysical properties due to failing to capture cloud vertical inhomogeneity are shown in Table 4. Cloud categories were described in Sect. 2. For the upper set of cases shown in the table, errors were not imposed and true cloud heights were used. In performing the retrieval the correct cloud base and top were used, but the cloud was assumed to be vertically homogeneous in terms of COD and phase; thus the cloud model is accurate for dense and diffuse clouds but not for inhomogeneous or liquid-topped clouds. This emulates a measurement where the cloud base and top are known from an

ancillary instrument such as a lidar. As expected, therefore, errors are similar for dense and diffuse clouds, which are vertically homogeneous and have known boundaries. For inhomogeneous clouds, which are thinner at the upper and lower edges, errors are slightly larger for $\tau_g$. The largest errors are found to be for liquid-topped clouds, particularly for $\tau_g$, for which errors are more than an order of magnitude larger. These errors are large because the cloud heights are effectively wrong for the liquid and ice layers of the cloud. A lidar that can classify phase would allow reduction of these errors down

to the level seen for other cloud types.

The middle set of cases shown in Table 4 includes imposed noise of 0.1 mW/(m$^2$ sr cm$^{-1}$) and temperature bias of 0.1 K, and uses retrieved cloud heights; the middle set uses true cloud heights while the final set uses retrieved cloud heights (error combinations e and f). These imposed errors are the same as those used to retrieve cloud height in Rowe et al (2016; see

Table 3). When the error is imposed, retrieval error increases only slightly for this level of imposed errors.

The final set of cases is identical to the middle set, but retrieved cloud heights are used discussed below, errors in cloud heights have the greatest effect on COD. Differences in retrieval errors for dense, diffuse, and inhomogeneous clouds are similar, as was also found to be the case for errors in cloud height (Rowe et al 2016, Table 3). Errors for liquid-topped

clouds are still greater than for other cloud types, but the difference is considerably smaller. Furthermore, retrieval errors for liquid-topped clouds are similar when true cloud heights are used but the cloud is assumed to be homogenously mixed as when the cloud height is retrieved. Trends for different combinations of imposed errors are similar (see Supplemental).
When retrieved cloud heights are used, the cloud is placed in the atmospheric model layer containing the effective cloud height retrieved with CO$_2$ slicing. This means that errors in COD are also strongly affected by the choice of atmospheric

layering. A more reasonable approach would be to base cloud thickness on a climatology.

Errors in retrieved microphysical properties due to assuming a spherical ice habit are shown in Table 5. The first column of the table shows the true ice habit. The upper set of data has no other imposed errors, while the lower two sets have the imposed errors given in the caption. The upper two sets use the true cloud height, while the lowest set uses retrieved cloud

height. Retrieval error in $r_{liq}$ is not shown because clouds were mainly ice. In the absence of imposed errors, compared to spheres, the increase in error is greatest for $\tau_g$, for which errors increase by an order of magnitude or more. The large increase in error in $\tau_g$ suggests that errors in habit mainly bias the magnitude rather than spectral shape of the cloud emissivity. However, in the presence of other imposed errors and when retrieved cloud heights are used, errors for non-spherical habits are less enhanced. Overall, errors are the smallest for solid columns (apart from spheres) and the largest for

hollow bullet rosettes and rough plates. Based on this sensitivity study, errors can be minimized by using a likely ice habit, although this is less important when cloud height is also retrieved.



## 6    Conclusions

This work introduces the CLoud and Atmospheric Radiation Retrieval Algorithm (CLARRA), which retrieves cloud effective height and microphysical properties (COD, ice fraction, effective radius of liquid, and effective radius of ice) from surface-based infrared downwelling radiance measurements made in Polar Regions. Cloud height retrievals use the $CO_2$ slicing method (Rowe et al 2016 and references therein). CLARRA rapidly retrieves first-guess cloud microphysical properties from a fast forward model that ignores scattering, and then uses the first-guess values as inputs into an optimal nonlinear inverse method using the Levenberg-Marquadt algorithm.


CLARRA was tested on 222 simulated spectra based on atmospheric and cloud conditions characteristic of the Arctic. Convergence was achieved in almost all cases, with a mean number of 4 iterations. Retrieval accuracy was found to be highest for a range of CODs between 0.04 and 5. Sensitivity to effective radius was found to be poor above about 50 µm, so retrieved values effective radii were bounded at this value. CLARRA was found to perform well when only model errors are
present: at a spectral resolution of 0.5 cm⁻¹ and for $\tau_g$ between 0.04 and 5 retrieved microphysical properties were unbiased, with errors (one standard deviation) of 0.01 ($\tau_g$), 0.04 ($f_{ice}$), 2 µm ($r_{liq}$) and 5 µm ($r_{ice}$).

Retrieval accuracy was tested for variety of imposed errors, including random noise as well as biases in the measured spectrum and atmospheric state. For example, at 0.5 cm⁻¹, for a combination of errors including noise of 0.2 RU, radiation
bias of -0.15 RU and PWV errors of 3%, errors (mean ± one standard deviation) are -0.02 ± 0.16 for $\tau_g$, 0.0 ± 0.2 for $f_{ice}$, -1 ± 4 µm for $r_{liq}$, and 1 ± 8 µm for $r_{ice}$. Errors in $\tau_g$ are most strongly affected by bias errors in cloud height. For example, for the same combination of imposed errors, if errors in cloud height exist due to retrieving cloud height (0.0 ± 0.3 km for clouds with bases below 2 km and -1.0 ± 0.7 km for clouds with bases above 2 km), errors in $\tau_g$ increase to 0.1 ± 0.3, but do not change much for other microphysical properties.


CLARRA includes a method for calculating gaseous transmission and emission terms at the effective instrument resolution, in order to improve computational efficiency. Forward model errors were found to be typically between within ± 0.3 RU, with median and 1st and 3rd quartile errors generally more than an order of magnitude smaller.

Comparison of the diagonals of the covariance matrix resulting from the inverse retrieval to actual retrieval errors suggested that the covariance matrix slightly underestimates errors at the 68% confidence interval; this is correctable by increasing model errors (~0.02 RU) and estimated measurement error due to noise (by ~25%). However, it should be noted that retrieval errors are still underestimated by the covariance matrix at higher confidence intervals (e.g. 95% and higher) owing to the non-Gaussian shape of the error distribution, which exhibits a slower-than Gaussian fall-off away from the strong
central peak. Furthermore, in the presence of bias errors, the covariance matrix was found to represent errors poorly, underscoring the importance of removing biases from measurements.

Sensitivity studies for vertically-varying clouds indicate that errors in retrieved microphysical properties are highest for liquid-topped clouds that are assumed to be homogeneously mixed-phase; furthermore, errors are only slightly higher when
cloud heights are retrieved from the spectra themselves, compared to using known cloud base and top. Errors in retrieved microphysical properties are similar for physically dense, diffuse, or vertically inhomogeneous clouds (having thinner upper



and lower boundaries), regardless of error. However, the gap in retrieved microphysical property error magnitudes between liquid-topped clouds and other cloud structures becomes smaller as imposed errors grow. Future work is needed to assess errors when multiple clouds are present. For different ice habits, sensitivity studies indicate that use of a reasonable guess
for the ice habit can help minimize errors.

Retrieval errors were found to be fairly constant up to about 2 cm$^{-1}$, after which they increase slightly with coarsening resolution. Retrieval accuracy at 2 cm$^{-1}$ is similar to accuracy at the resolution of the AERI instrument (0.5 cm$^{-1}$), which has been used for cloud microphysical property retrievals at a variety of polar locations. This indicates that a resolution of 2 cm$^{-1}$
is sufficient for accurate cloud property retrievals using a surface-based infrared spectrometer. For the coarser resolution of 4 cm$^{-1}$, for the base set of 222 clouds and for errors expected to be characteristic of reanalysis data (temperature bias of 0.2 K and water vapour bias of 3%), for an instrument radiation bias of 0.2 mW/(m$^2$ sr cm$^{-1}$) and noise level of 0.2 mW/(m$^2$ sr cm$^{-1}$), retrieval errors were found to be $0.1 \pm 0.6$ for optical depth, $0.0 \pm 0.3$ for ice fraction, $0 \pm 2$ μm for effective radius of liquid, and $2 \pm 2$ μm for effective radius of ice. These results indicate the potential utility of a moderately low resolution,
portable, surface-based infrared spectrometer for providing microphysical properties that can help constrain climate models.

## 7    Code and data availability

Simulated radiances at monochromatic resolution (Cox et al 2015) are available on the Arctic Observing Network (AON) Arctic data repository (https://arcticdata.io/catalog/#view/doi:10.5065/D61J97TT); updated versions with errors corrected are available by email to the corresponding author. Computer code is available at Bitbucket, including repositories
containing Python computer code (runDisort_py; https://bitbucket.org/clarragroup/rundisort_py/src/master/) and Matlab/Octave computer code (runDisort_mat; https://bitbucket.org/clarragroup/rundisort_mat/src/master/) for creating cloudy-sky spectra using DISORT (Stamnes et al 1988). See also Rowe et al (2013; 2016).

## 8    Author contribution

Steven Neshyba calculated Legendre moments and single-scatter albedo from single-scattering parameters. Christopher Cox
led creation of simulated spectra used in this work. Von Walden conceived of the idea and provided guidance. Penny Rowe performed all other calculations and wrote the manuscript with input from all authors.

**Acknowledgements**

This work was funded by National Science Foundation grants ARC-1108451 and PLR 1543236. S. N. acknowledges NSF
Grant CHE 1807898.

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





Table 1. Statistics of cloud microphysical properties. Standard deviations were calculated for the logarithms of cloud optical depth referenced to the geometric limit ($\tau_g$), effective radius of liquid ($r_{liq}$) and effective radius of ice ($r_{ice}$) because distributions for the logarithms were found to be more Gaussian in shape; these standard deviations were converted positive and negative standard deviations for these quantities. The ice fraction, $f_{ice}$, peaks strongly at both limits; thus no standard deviation is provided.

| Quantity | Units | Mean | Std. dev. | range |
|---|---|---|---|---|
| $\tau_g$ | (tau) | 2 | -0.5, +2 | $0.03 - 9.3$ |
| $f_{ice}$ | (fract) | 0.5 | - | 0 - 1 |
| $r_{liq}$ | (µm) | 10 | -3, +4 | 2 - 21 |
| $r_{ice}$ | (µm) | 25 | -9, +14 | 5 - 58 |

Table 2. Errors in retrieved cloud microphysical properties for base set of spectra due to model error only (no errors imposed), for spectral resolutions indicated. Errors are shown for cloud geometric optical depth ($\tau_g$), ice fraction ($f_{ice}$) effective radius of liquid ($r_{liq}$), and effective radius of ice ($r_{ice}$), for four ranges in $\tau_g$.

| | | 0.1 cm$^{-1}$ | 0.5 cm$^{-1}$ | 4 cm$^{-1}$ |
|---|---|---|---|---|
| $\tau_g$ | $\tau_g < 0.25$ | $0.000 \pm 0.003$ | $-0.001 \pm 0.006$ | $0.001 \pm 0.003$ |
| | $0.25 < \tau_g < 0.4$ | $0.003 \pm 0.005$ | $0.003 \pm 0.004$ | $0.004 \pm 0.005$ |
| | $0.4 < \tau_g < 5$ | $0.000 \pm 0.012$ | $-0.000 \pm 0.013$ | $0.001 \pm 0.016$ |
| | $\tau_g > 5$ | $0.1 \pm 0.3$ | $0.1 \pm 0.3$ | $0.2 \pm 0.3$ |
| $f_{ice}$ | $\tau_g < 0.25$ | $-0.03 \pm 0.10$ | $-0.03 \pm 0.10$ | $0.04 \pm 0.11$ |
| | $0.25 < \tau_g < 0.4$ | $-0.02 \pm 0.06$ | $-0.03 \pm 0.08$ | $0.01 \pm 0.06$ |
| | $0.4 < \tau_g < 5$ | $-0.00 \pm 0.03$ | $0.00 \pm 0.04$ | $0.02 \pm 0.11$ |
| | $\tau_g > 5$ | $0.003 \pm 0.014$ | $0.00 \pm 0.01$ | $-0.05 \pm 0.10$ |
| $r_{liq}$ | $\tau_g < 0.25$ | $1 \pm 3$ | $-0 \pm 5$ | $-1 \pm 3$ |
| (µm) | $0.25 < \tau_g < 0.4$ | $-0.5 \pm 1.3$ | $-0.2 \pm 1.8$ | $-1.4 \pm 1.9$ |
| | $0.4 < \tau_g < 5$ | $0.0 \pm 0.5$ | $0.0 \pm 0.9$ | $-0.2 \pm 1.2$ |
| | $\tau_g > 5$ | $-0.3 \pm 0.7$ | $-0.2 \pm 0.5$ | $-0.4 \pm 1.3$ |
| $r_{ice}$ | $\tau_g < 0.25$ | $1 \pm 6$ | $2 \pm 7$ | $2 \pm 5$ |
| (µm) | $0.25 < \tau_g < 0.4$ | $5 \pm 6$ | $5 \pm 5$ | $5 \pm 6$ |
| | $0.4 < \tau_g < 5$ | $1 \pm 3$ | $0 \pm 4$ | $1 \pm 4$ |
| | $\tau_g > 5$ | $1 \pm 2$ | $1 \pm 2$ | $5 \pm 7$ |



Table 3. Errors in retrieved cloud microphysical properties (where COD refers to cloud optical depth in the geometric limit, $r_{liq}$ and $r_{ice}$ are the effective radii of liquid and ice, and SD indicates standard deviation) for base set spectra at a resolution of 0.5 cm$^{-1}$ and optical depth between 0.4 and 5, with imposed errors on measurement and atmospheric state. Combined errors are described in the text. For PWV biases of 10% and -10% and set d of combined errors, 12, 9, and 2 points (respectively) were omitted due to having anomalously large errors.

| | COD | | Ice fraction | | $r_{liq}$ (µm) | | $r_{liq}$ (µm) | |
|---|---|---|---|---|---|---|---|---|
| | Mean | SD | Mean | SD | Mean | SD | Mean | SD |
| None | 0.002 | 0.007 | -0.01 | 0.09 | 0.0 | 0.7 | 0 | 4 |
| Noise (0.2 RU) | 0.00 | 0.04 | 0.01 | 0.12 | 0 | 3 | 1 | 9 |
| Bias (0.2 RU) | -0.03 | 0.03 | 0.05 | 0.14 | -1 | 2 | -3 | 7 |
| Bias (-0.2 RU) | 0.03 | 0.03 | -0.05 | 0.14 | 0 | 3 | 3 | 5 |
| Temp. (0.2 K) | 0.0 | 0.02 | 0.00 | 0.08 | 0 | 2 | 0 | 6 |
| Temp. (-0.2 K) | 0.0 | 0.01 | 0.01 | 0.09 | 0 | 2 | -1 | 5 |
| PWV (3%) | 0.01 | 0.03 | -0.08 | 0.13 | 0 | 2 | 2 | 7 |
| PWV (-3%) | -0.01 | 0.03 | 0.10 | 0.16 | -1 | 2 | -5 | 9 |
| PWV (10%) | 0.05 | 0.16 | -0.1 | 0.2 | -1 | 5 | 1 | 11 |
| PWV (-10%) | -0.04 | 0.14 | 0.2 | 0.3 | -1 | 3 | -5 | 10 |
| CO2 Slicing | 0.1 | 0.4 | -0.03 | 0.12 | 0 | 3 | 1 | 5 |
| Combined, a | -0.02 | 0.16 | -0.0 | 0.2 | -1 | 4 | 1 | 8 |
| Combined, b | 0.03 | 0.18 | 0.1 | 0.3 | -1 | 3 | 0 | 9 |
| Combined, c | 0.1 | 0.3 | -0.1 | 0.2 | 0 | 4 | 2 | 9 |
| Combined, d | 0.1 | 0.5 | 0.1 | 0.3 | 0 | 4 | -2 | 9 |



Table 4. Errors in retrieved cloud microphysical properties for macroscopically varying clouds at a resolution of 0.5 cm$^{-1}$ (where COD refers to cloud optical depth in the geometric limit, $r_{liq}$ and $r_{ice}$ are the effective radii of liquid and ice, and SD indicates standard deviation). For the upper set of cases, errors were not imposed (error = n) and true cloud heights were used. The middle set of cases includes imposed noise of 0.1 mW/(m$^2$ sr cm$^{-1}$) and temperature bias of 0.1 K, with true cloud heights (error = y); for the lowest set, imposed errors are identical except that retrieved cloud heights were used (error = y*).

| Cloud type | Error | COD | | Ice fraction | | $r_{liq}$ (µm) | | $r_{ice}$ (µm) | |
|---|---|---|---|---|---|---|---|---|---|
| | | Mean | SD | Mean | SD | Mean | SD | Mean | SD |
| Dense | n | 0.004 | 0.012 | -0.01 | 0.01 | 0.0 | 0.4 | 2 | 6 |
| Diffuse | n | 0.004 | 0.012 | 0.00 | 0.02 | -0.1 | 0.5 | 2 | 6 |
| Inhomogeneous | n | 0.009 | 0.016 | -0.01 | 0.02 | 0.0 | 0.4 | 2 | 6 |
| Liquid topped | n | 0.06 | 0.06 | -0.05 | 0.08 | 0.0 | 1.1 | 5 | 9 |
| Dense | y | 0.011 | 0.016 | -0.03 | 0.06 | 0.0 | 0.7 | 2 | 6 |
| Diffuse | y | 0.003 | 0.011 | 0.00 | 0.01 | 0.0 | 0.4 | 2 | 6 |
| Inhomogeneous | y | 0.009 | 0.016 | -0.01 | 0.02 | 0.1 | 0.5 | 2 | 6 |
| Liquid topped | y | 0.06 | 0.07 | -0.05 | 0.08 | 0.0 | 1.1 | 6 | 9 |
| Dense | y* | 0.01 | 0.04 | 0.00 | 0.06 | -0.1 | 1.6 | 2 | 8 |
| Diffuse | y* | 0.05 | 0.08 | -0.01 | 0.08 | 0.0 | 1.2 | 3 | 6 |
| Inhomogeneous | y* | 0.03 | 0.08 | -0.02 | 0.05 | 0.1 | 1.5 | 3 | 7 |
| Liquid topped | y* | 0.07 | 0.09 | -0.02 | 0.07 | -0.3 | 1.7 | 6 | 8 |






Table 5. Errors cloud microphysical properties retrieved assuming a spherical ice habit, for ice clouds of varying habit (first column). COD refers to cloud optical depth in the geometric limit, $r_{ice}$ is the effective radius of ice, and SD indicates standard deviation. For the upper set of cases, errors were not imposed (error = n) and true cloud heights were used. The middle set of cases includes imposed noise of 0.1 mW/(m$^2$ sr cm$^{-1}$) and temperature bias of 0.1 K, with true cloud heights (error = y); for the lowest set, imposed errors are identical except that retrieved cloud heights were used (error = y*). The final column shows the number of cases omitted due to having rms differences of greater than 1 RU between the measured and retrieved radiance.

| Habit | Error | COD Mean | SD | Ice fraction Mean | SD | $r_{ice}$ (μm) Mean | SD | Omit |
|---|---|---|---|---|---|---|---|---|
| Sphere | n | 0.01 | 0.02 | 0.000 | 0.010 | 1 | 3 | 0 |
| Hollow bullet rosette | n | 0.6 | 0.4 | 0.04 | 0.06 | 8 | 6 | 2 |
| Smooth solid column | n | 0.3 | 0.2 | 0.02 | 0.03 | 5 | 5 | 0 |
| Rough solid column | n | 0.3 | 0.2 | 0.02 | 0.03 | 5 | 5 | 0 |
| Smooth plate | n | 0.5 | 0.3 | -0.01 | 0.05 | 7 | 3 | 4 |
| Rough plate | n | 0.5 | 0.3 | 0.05 | 0.06 | 6 | 3 | 2 |
| Sphere | y | 0.02 | 0.02 | 0.05 | 0.06 | 1 | 4 | 0 |
| Hollow bullet rosette | y | 0.6 | 0.4 | 0.04 | 0.08 | 9 | 8 | 2 |
| Smooth solid column | y | 0.3 | 0.2 | 0.000 | 0.010 | 5 | 6 | 0 |
| Rough solid column | y | 0.3 | 0.2 | 0.03 | 0.05 | 5 | 6 | 0 |
| Smooth plate | y | 0.5 | 0.3 | 0.00 | 0.03 | 7 | 3 | 4 |
| Rough plate | y | 0.6 | 0.4 | 0.02 | 0.03 | 6 | 3 | 3 |
| Sphere | y* | -0.1 | 0.4 | 0.10 | 0.16 | 3 | 5 | 1 |
| Hollow bullet rosette | y* | 0.5 | 0.3 | 0.11 | 0.10 | 9 | 7 | 3 |
| Smooth solid column | y* | 0.3 | 0.2 | 0.10 | 0.07 | 4 | 5 | 0 |
| Rough solid column | y* | 0.3 | 0.2 | 0.10 | 0.08 | 4 | 5 | 0 |
| Smooth plate | y* | 0.5 | 0.3 | 0.07 | 0.14 | 7 | 3 | 4 |
| Rough plate | y* | 0.6 | 0.3 | 0.14 | 0.13 | 6 | 3 | 3 |




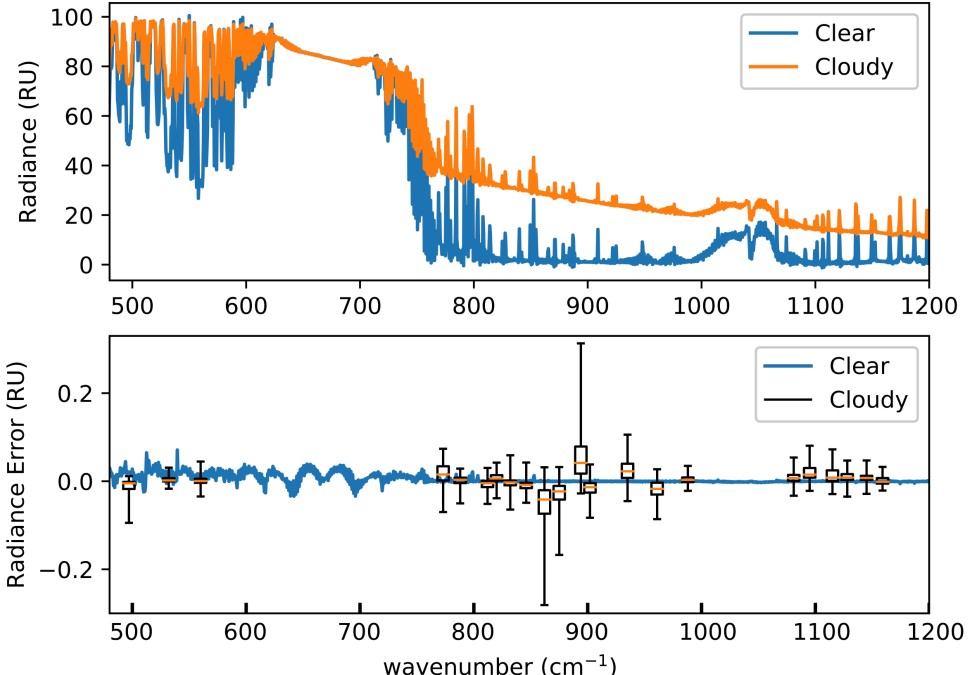

Figure 1. Clear and cloud-sky downwelling radiance (1 RU = mW / [m$^2$ sr cm$^{-1}$]) for typical polar atmosphere at a resolution of 0.5 cm$^{-1}$ (top). Model errors (model – true) in downwelling radiances for the clear-sky radiance shown in the top panel, and box and whiskers plots of model errors for all radiances, averaged in microwindows (horizontal lines give the median, boxes give the 1$^{st}$ and 3$^{rd}$ quartiles, and whiskers give the range; bottom).





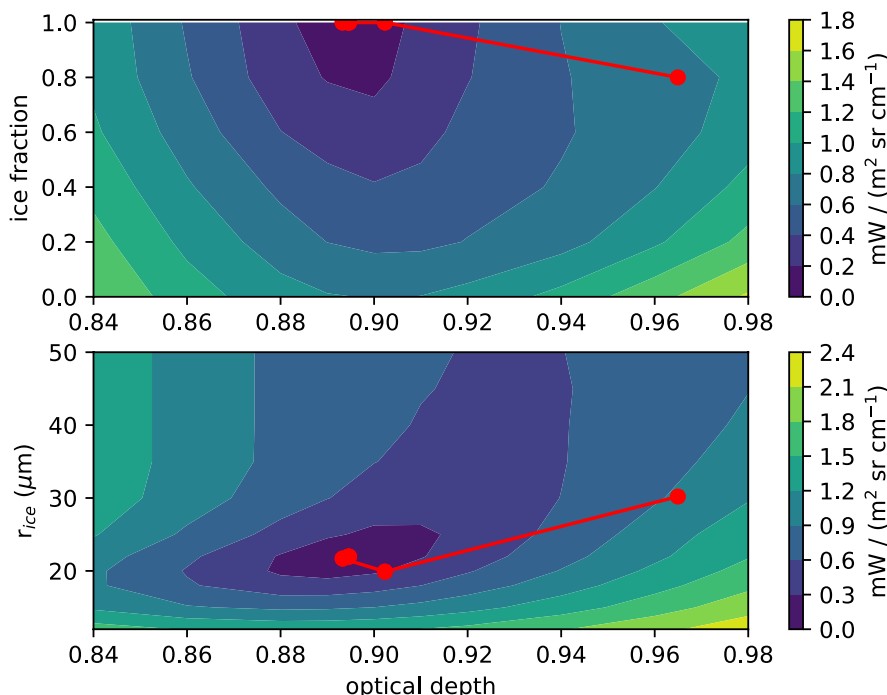

Figure 2. Error contours for retrievals of ice effective radius ($r_{ice}$), ice fraction, and cloud optical depth, as rms error in radiances for an ice-only cloud. The retrieval trajectory (red line) and results for each iteration (red circles) is superimposed on the contour surface.




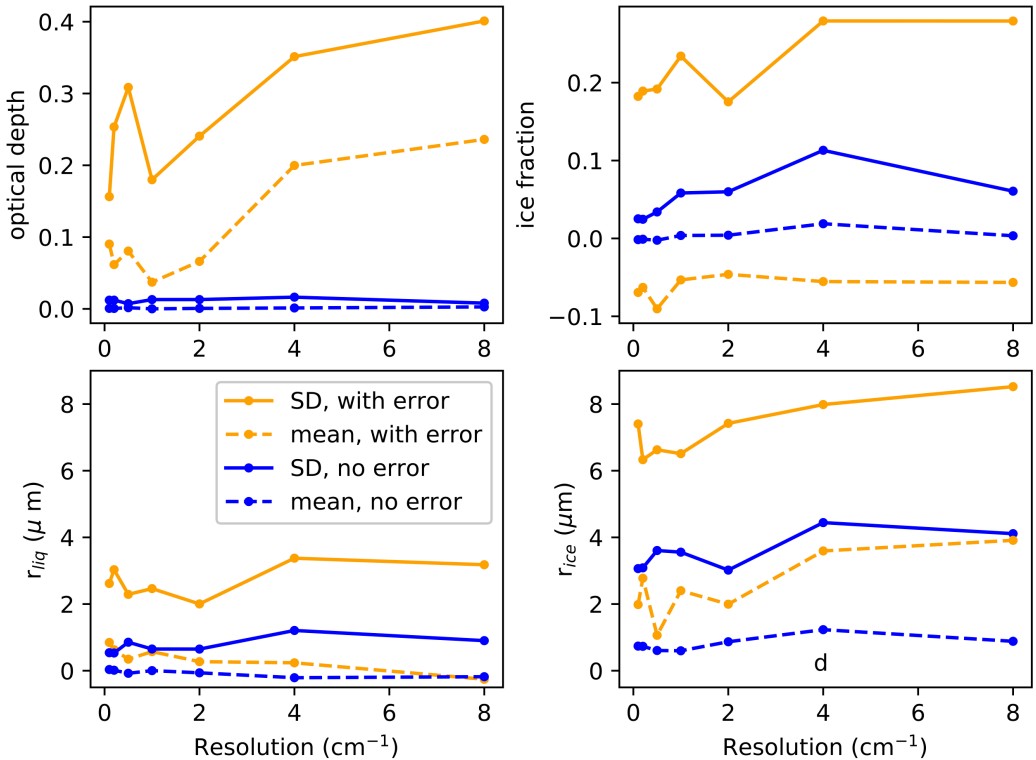

Figure 3. Errors in retrieved microphysical cloud properties (mean and standard deviation; SD) as a function of resolution, where $r_{liq}$ is the effective radius of liquid and $r_{ice}$ is the effective radius of ice, for cases with no imposed error (no error) and with imposed errors as described in the text.