# Peer review of "Toward autonomous surface-based infrared remote sensing of polar clouds: Retrievals of cloud optical and microphysical properties"

_Atmospheric Measurement Techniques, 2019_

## Referee Comment (RC1) · D. D. Turner (Referee) · 9 Jul 2019

This paper describes a cloud retrieval algorithm (CLARRA) that is applied to synthetically generated radiance data to provide a dataset of several hundred clouds (single phase ice, single phase liquid, mixed-phase, etc). The objective of this work, as stated in the abstract, is to show how the bias in the retrieved cloud properties changes as the spectral resolution of the ground-based instrument used to provide the radiance observations becomes coarser. The overall goal is to show that instruments with relatively coarse spectral resolution (e.g., 8 cm-1) yield more-or-less similar results as a high spectral resolution (e.g., 0.1 cm-1) instrument.

[Figure]

I have a multitude of concerns with this work, and believe that the paper should not be published in its current form.

My primary one is that the "new" CLARRA algorithm is almost exactly the same as the so-called "MIXCRA" algorithm of Turner (J Appl Meteor, 2005). In fact, it is almost like the authors were trying to disguise this as there is not a single reference to the MIXCRA paper in sections 1, 2, 3.1, or 3.2; yet it is clear the authors know about MIXCRA because it is referenced at the top of section 3.3. It is not clear what makes CLARRA different from MIXCRA.

My second primary concern is simply: Is this paper really adding any new knowledge? I presume that they are using observations in spectral regions that are between absorption lines (these are often called "microwindows"), and the cloud properties in these microwindows are essentially unchanged. This allows the radiance to be averaged from higher spectral resolution to that of the width of the microwindow; this was done in the MIXCRA paper (see table 1 in that paper). Indeed, the microwindows used in the MIXCRA paper are between 2 and 8 cm-1 wide, depending on the spectral region. So while the MIXCRA paper didn't specifically test retrievals performed at 0.5 vs. 4 cm-1, it already is showing the impact. [As a side note: the authors really do need to include a table on what microwindows are being used in this study. In addition to the actual microwindows, if the microwindow is actually only 4 cm-1 wide, do the authors limit their spectral width of that "channel" to only or do they include the absorption lines that exist on the sides of these microwindows? If they are testing 0.1 cm-1 resolution, do they allow multiple "channels" within a given microwindow, or only a single channel? Without this information, the results here cannot be reproduced.]

The uncertainties assumed for temperature and humidity profiles in the reanalysis (around line 355) are shockingly small. These uncertainties might be true for a large average, but in a scene-by-scene way the errors will be much, much larger. For example, if the reanalysis believes that the sky is cloud-free above the instrument, the reanalysis may have developed a surface-based inversion that would not be there in

reality because of the cloud. An example of how the presence of a cloud modifies the temperature profile beneath the cloud is given by Miller et al. JGR 2013 (which includes Walden as a coauthor). Because clouds are so hard to represent properly in large-scale models, I think the authors need to use more representative uncertainties (e.g., many degrees for temperature, and at least 20% for water vapor) for the temperature and humidity profiles, and show the impact of these uncertainties.

The authors are using the far-infrared for several of their channels; indeed, using those channels are really important for discriminating the cloud phase (see Turner et al. JAM 2003 as well as Turner JAM 2005). However, water vapor absorbs strongly in the far-infrared, and if the PWV is large enough the window will be opaque purely due to water vapor absorption. Thus, the range of cloud optical depth that can be sensed depends strongly on the PWV; this needs to be discussed in this paper.

I think that the authors have missed a real opportunity to talk about how the uncertainties in the retrieved products covaries (i.e., by looking at the off-diagonal elements of the posterior covariance matrix). This was one of the shortcomings of the MIXCRA paper, and expansion of that here would add some new insights to the community. For example, as the cloud emissivity moves towards unity, there will likely be a high amount of correlated error between Reff,ice and Reff,liq. Ditto when the cloud emissivity is small. How does the ice fraction uncertainty covary with the other retrieved variables, especially in different areas of the solution space?

A different question along the same lines as above is this: does the accuracy and covariance between the cloud properties change if the retrieval is configured to retrieve (total_tau, ice_fraction, Reff_ice, Reff_liq) vs. (tau_liq, tau_ice, Reff_ice, Reff_liq)? The MIXCRA algorithm was initially the first (in JAM 2005), but was changed to the latter (Turner and Eloranta TGRS 2008) and showed pretty good results relative to the HRSL during MPACE.

The authors really didn't spend any time discussing the different technologies that could

be used to provide these radiance observations, or why they would be "cheaper" from the instrument that they assumed (which seems to be the AERI). Radiometers using a finite number of channels with bolometers as detectors are one possibility; there are many papers by the so-called "TICFIRE" project being run out of the Canadian Space Agency that might be useful. But how important is the calibration of cheaper systems like this? The authors did show results if the radiance bias was 0.2 RU, but that is a pretty small error – a more realistic error might be 1 or 2 RU. Do these results scale? [For example, even with a carefully calibrated AERI, radiance biases close to 1 RU have been reported – see Delamere et al. JGR 2010. It is hard to imagine that a cheaper radiometer would have better spectral calibration than the AERI.]

Some more minor points: • How many streams are being used in DISORT? Fewer streams make the RT code faster, but will decrease the accuracy. • Eq 18 is incorrect if gamma > 0. The correct formulation was originally provided by Masiello et al. QJRMS 2012, but was also presented in Turner and Löhnert JAMC 2014. • "the ideal range for tau is between 0.4 and 5" (line 425). The authors really should indicate how often this is expected to happen in the Arctic. The Cox et al. JAMC 2014 paper provides some information on this, at least for that site. • Is there a minimum number of spectral microwindows that need to be used? Stated a different way, how do the errors in the retrieved cloud properties change for different microwindow subsets?

I would be very happy to talk with the authors about this review and my thoughts to how this paper could be improved, should they want to do so.

Sincerely, Dave Turner
* * *

---

## Referee Comment (RC2) · Bryan A. Baum (Referee) · 9 Jul 2019

Bryan A. Baum (Referee)

baum@stcnet.com

This manuscript builds on earlier work (Rowe et al. 2016; cloud height retrieval) to include the inference of cloud optical and microphysical properties (optical thickness and particle size), which also requires cloud thermodynamic phase. The current methodology now includes consideration of cloud phase, vertical inhomogeneity, ice habit, sensor noise, and more, in an optimal estimation framework. Good use is made of the ice particle single-scattering property database of Yang et al. (2013). Their interest is the examination of the retrieval sensitivity to spectral resolution from 0.1 to 8 cm-1. As noted in the Abstract, the retrieval accuracy is basically unaffected by resolution from

0.1 to 2 cm-1 and decreases slightly as the resolution increases thereafter. The stated goal is to work towards autonomous surface-based IR remote sensing of polar cloud properties by a portable spectrometer system.

The methodology is clearly stated, adequate references are made to work by earlier studies although more recent articles may be available, and the analyses are straight-forward. I am glad to see that the software is being made available to the community as described in Section 7. My view is that the paper is suitable for publication pending relatively minor revisions that address the comments that follow.

If there is one primary suggestion to offer, it would be to compare the cloud properties obtained by this method to those obtained from CALIPSO, where new Version 4 products are now available (or to a coincident ground-based lidar if possible). Of particular note is that the V4 products have significant improvements in calibration and cloud/aerosol properties. There will be differences between satellite- and surface-based products that will bear further investigation, but this may be outside the scope of this particular study. CALIPSO cloud products have been used heavily in the development and testing phase of many satellite-based cloud retrieval efforts, especially with the discrimination of cloud thermodynamic phase which is a critical component of the current study. As noted in Section 5.4, imperfect cloud phase discrimination can greatly increase the retrieval errors (lines 500-505).

General comments:

Lines 58-62: Two points to suggest here: 1. The authors point out the need for portable, low-cost, autonomous IR spectrometers that can make continuous measurements. But these measurements complement those from polar-orbiting IR spectrometers including IASI (on Metop-A/B/C), AIRS, and CrIS. It would be useful to provide an example where the surface measurements fill the gaps between satellite overpasses. 2. Additionally, a primary benefit to surface-based measurements is that the boundary layer is much better characterized than with profiles inferred from a satellite-based spectrometer. In
particular, my impression is that the boundary layer profiles are much improved when temperature inversions are present, and this will impact the cloud properties if the layer is at/below the inversion.

Section 3: This is a long section (over 6 pages) that discusses the Cloud and Atmospheric Radiation Retrieval Algorithm (CLARRA) in quite a bit of detail. Cloud height was discussed in great detail in Rowe et al. (2016) and is not repeated herein. Perhaps the readability would be improved by moving much of the theoretical development into an Appendix.

Minor comments:

Line 19: please define exactly what is meant by "mixed phase" - is it a homogeneous mixture of ice and liquid particles or something else?

The word "infrared" appears 26 times in the paper - could contract to IR

Line 47: include more up-to-date L'Ecuyer papers, e.g., "Reassessing the effect of cloud type on Earth's energy balance in the age of active spaceborne observations. Part I: Top-of-atmosphere and surface", by TS L'Ecuyer, Y Hang, AV Matus, Z Wang, in Journal of Climate, 2019. There is also a Part 2 manuscript in review.

Lines 293-294: Radiances are selected in two bands: 400 to 600 cm-1 and from 750 to 1300 cm-1. As the method is using selected wavenumbers for each chosen spectral resolution, it would be useful to state them in this paper rather than in the supplemental.

Lines 340; 352; 557: suggest changing "in order to" to "to"

Line 356: change "found such error" to "found such errors"

Lines 503-505: Cloud height: is $CO_2$ slicing used for both water and ice clouds? If so....might want to change this so it's used primarily for ice clouds and use 11-$\mu$m for optically thick clouds.

Line 536: Polar Regions does not have to be capitalized.

Line 561: suggest changing "correctable" to "mitigated"

---

## Author Comment (AC1) · 20 Aug 2019

Reply to Referee 1. We indicate a referee comment with "Referee" and our response with "Authors".

Referee: I have a multitude of concerns with this work, and believe that the paper should not be published in its current form. My primary one is that the "new" CLARRA algorithm is almost exactly the same as the so-called "MIXCRA" algorithm of Turner (J Appl Meteor, 2005). In fact, it is almost like the authors were trying to disguise this as there is not a single reference to the MIXCRA paper in sections 1, 2, 3.1, or 3.2; yet it is clear the authors know about MIXCRA because it is referenced at the top of section

3.3. It is not clear what makes CLARRA different from MIXCRA.

Authors: Our main purpose is to inform instrument development by quantifying the effect of instrument resolution on cloud property retrievals, which to our knowledge is novel and within the scope of AMT. Furthermore, aspects of CLARRA are novel, including formulations for matching instrument resolution for the cloud property retrievals, as described in Section 3.1. Other novel aspects of this work include testing retrievals on a wide set of simulations that are meant to be characteristic of the annual cycle of Arctic clouds and atmosphere (based on previous work by the authors; including different ice habits and vertically-varying clouds; Section 2) and quantifying the effect on the retrieval of a range of errors in the instrument and atmospheric state (Section 4). We have modified the paper throughout to make the purpose clearer.

Given our main purpose, we do not see similarities between CLARRA and published work as a source of concern. The optimal nonlinear inverse method in CLARRA is similar to MIXCRA (Turner 2005) and to other work, as cited in the submitted manuscript: "The inverse method is similar to the method of Turner (2005) which has been used to retrieve cloud properties from AERI instruments in the Arctic (Turner 2005; Cox et al 2014). Similar inverse methods have also been used from satellite instruments (L'Ecuyer et al 2006; Wang et al 2006)." To make this clearer, we have combined these sentences and added references to Poulsen et al., 2012 (and L'Ecuyer et al, 2019, in response to reviewer 2). Also, it should be noted that polar cloud property retrievals in general build on a large body of published work, including work by the authors (Mahesh et al 2001; Rathke et al 2002a,b).

We do not see the relevance of the MIXCRA paper to Section 2, which is based on Cox et al, 2016, or to Sections 3.1 or 3.2, which we believe are largely novel to our work (note that the revised manuscript is restructured somewhat following Reveiewer 2). However, we realize that our results and discussion section would benefit from comparison to published work by Turner 2005 and others. Therefore we will add text such as the following to Section 5.1:
[Figure]

"We find that the retrievals lose sensitivity to COD between about 4 and 10 (see Sect. 5.2 below); in previous work retrieving cloud properties from downwelling IR radiances in a similar wavenumber range, cut-offs of 4 to 6 were used (Mahesh et al 2001; Rathke 2002a,b; Turner 2005). Here, we constrain the retrievals to an optical depth of 10 and focus primarily on results for optical depths <= 4. The retrievals were found to lose sensitivity to effective radius above about 50 $\mu$m (see Supplemental), which is in keeping with Rathke and Fischer (2000) and Garrett and Zhao (2013), but differs from the cut-off values of 25 $\mu$m used by Mahesh et al (2001) and of 100 $\mu$m by Turner (2005)."

Because CLARRA has similarities to a variety of algorithms, it is beyond the scope of this work to detail them all. However, we provide sufficient detail to determine differences by comparing our paper to the published literature. For example, differences in CLARRA relative to MIXCRA include: 1) CLARRA includes a cloud height retrieval (Section 3; the revised manuscript also mentions this in the introduction). 2) CLARRA includes a forward retrieval that is computationally fast (Section 3.2). 3) CLARRA accounts for gaseous emission at instrument resolution using a novel method (convolving radiances and transmittances with the instrument lineshape; Section 3.1 in the submitted manuscript). 4) The optimal nonlinear inverse method in CLARRA uses the Levenberg-Marquardt formulation, and uses the radiance as the observation (the variable y in Rodgers 2000; R in Eqn (16) in the submitted manuscript). 5) CLARRA uses temperature-dependent liquid water refractive indices (Section 3.2). 6) CLARRA is written primarily in Python and calls DISORT directly from Python via f2py.

Referee: My second primary concern is simply: Is this paper really adding any new knowledge? I presume that they are using observations in spectral regions that are between absorption lines (these are often called "microwindows"), and the cloud properties in these microwindows are essentially unchanged. This allows the radiance to be averaged from higher spectral resolution to that of the width of the microwindow; this was done in the MIXCRA paper (see Table 1 in that paper). Indeed, the microwindows

used in the MIXCRA paper are between 2 and 8 cm-1 wide, depending on the spectral region. So while the MIXCRA paper didn't specifically test retrievals performed at 0.5 vs. 4 cm-1, it already is showing the impact.

Authors: We understand the point that averaging over microwindows implies that resolution is not very important, but the issue is more complex because strong gaseous emission lines bleed into the microwindows at coarser resolution, so that there is still a need to quantify the effect of resolution. The reason microwindows are used is to minimize the contribution of emission by gases. Gaseous emission lowers sensitivity to cloud and enhances errors. At a resolution of 0.1 cm-1, in a microwindow of 4 cm-1, the contribution from gaseous absorption lines outside the microwindow will be minimal. However, as resolution gets coarser, the gaseous absorption lines bordering the microwindow contribute more and more, decreasing sensitivity and increasing errors. To design a lower-resolution instrument, it is essential to quantify these errors to determine at what resolution they become important. We have added text similar to that above to Section 3.1.

Furthermore, resolution affects the accuracy of cloud height retrievals, which in turn affects the accuracy of microphysical cloud property retrievals, as discussed in the manuscript. Thus, our paper adds new knowledge by quantifying how cloud-property retrievals depend on resolution, from 0.1 to 8 cm-1. To make this more clear, we increased the resolution range to 20 cm-1 and included more discussion of how cloud-height retrieval accuracy affects the accuracy of microphysical cloud property retrievals.

Referee: [As a side note: the authors really do need to include a table on what microwindows are being used in this study.

Authors: A table showing the microwindows was included in the Supplemental and discussed in Section 3: "Selected microwindows (see Supplemental) are similar to those in Turner (2005) ..." We have moved the table into the main text, added a second set of microwindows (see the response to the final comment), and added the relevant

references.

Referee: In addition to the actual microwindows, if the microwindow is actually only 4 cm-1 wide, do the authors limit their spectral width of that "channel" to only or do they include the absorption lines that exist on the sides of these microwindows? If they are testing 0.1 cm-1 resolution, do they allow multiple "channels" within a given microwindow, or only a single channel? Without this information, the results here cannot be reproduced.]

Authors: Section 3.1 described calculation of radiances and transmittances (Eqns 1-8) and states that they are then averaged within microwindows; these are then used to compute effective-resolution optical depths (Eqns 9-10). For the observations (which here are simulations), radiances are averaged in the microwindows. Using the equations and microwindows provided, our results should be reproducible. (Note that much of Sect. 3.1 was moved to the Appendix, following the suggestions of Reviewer 2; we also made changes for clarity).

Referee: The uncertainties assumed for temperature and humidity profiles in the reanalysis (around line 355) are shockingly small. These uncertainties might be true for a large average, but in a scene-by-scene way the errors will be much, much larger. For example, if the reanalysis believes that the sky is cloud-free above the instrument, there analysis may have developed a surface-based inversion that would not be there in reality because of the cloud. An example of how the presence of a cloud modifies the temperature profile beneath the cloud is given by Miller et al. JGR 2013 (which includes Walden as a coauthor). Because clouds are so hard to represent properly in large-scale models, I think the authors need to use more representative uncertainties (e.g., many degrees for temperature, and at least 20% for water vapor) for the temperature and humidity profiles, and show the impact of these uncertainties.

Authors: We believe use of the mean error is reasonable. According to Wesslen et al (2014) the 95% confidence interval is within 0.3 K for temperature and within a few

percent for humidity. Also, as noted in the text, bias errors in humidity or temperature give similar results as considerably larger errors that vary in sign with height, which is likely for reanalysis data (Wesslen et al, 2014). Finally, note that in addition to the mean errors, we tested the effect on the retrieval of errors in water vapor as high as a 10% bias at all heights. We have added retrievals for a temperature bias of 1 K, increased error magnitudes in combinations of errors, and changed "typical errors expected" to "biases" in the abstract. (Our main conclusions are unaffected).

Referee: The authors are using the far-infrared for several of their channels; indeed, using those channels are really important for discriminating the cloud phase (see Turner et al. JAM2003 as well as Turner JAM 2005).

Authors: Yes. This point was published prior to these papers by Rathke et al (2002), whom we reference:

Rathke, C., Fischer, J., Neshyba, S., & Shupe, M. (2002). Improving IR cloud phase determination with 20 microns spectral observations. Geophysical Research Letters, 29(8), 50–1–50–4. http://doi.org/10.1029/2001GL014594.

Referee: However, water vapor absorbs strongly in the far-infrared, and if the PWV is large enough the window will be opaque purely due to water vapor absorption. Thus, the range of cloud optical depth that can be sensed depends strongly on the PWV; this needs to be discussed in this paper.

Authors: This is a good point. We have changed Section 2 to clarify that the range of PWV varied from 0.2 to 3 cm, which previous work has shown is appropriate for polar regions. For this range of PWV, the far IR window was not opaque.

Referee: I think that the authors have missed a real opportunity to talk about how the uncertainties in the retrieved products covaries (i.e., by looking at the off-diagonal elements of the posterior covariance matrix). This was one of the shortcomings of the MIXCRA paper, and expansion of that here would add some new insights to the

community. For example, as the cloud emissivity moves towards unity, there will likely be a high amount of correlated error between Reff,ice and Reff,liq. Ditto when the cloud emissivity is small. How does the ice fraction uncertainty covary with the other retrieved variables, especially in different areas of the solution space? A different question along the same lines as above is this: does the accuracy and covariance between the cloud properties change if the retrieval is configured to retrieve (total_tau, ice_fraction, Reff_ice, Reff_liq) vs. (tau_liq, tau_ice, Reff_ice, Reff_liq)? The MIXCRA algorithm was initially the first (in JAM 2005), but was changed to the latter (Turner and Eloranta TGRS 2008) and showed pretty good results relative to the HRSL during MPACE.

Authors: We agree that this would be interesting. However, it is beyond the scope of the current paper. We will look into this in future work.

Referee: The authors really didn't spend any time discussing the different technologies that could be used to provide these radiance observations, or why they would be "cheaper" from the instrument that they assumed (which seems to be the AERI). Radiometers using a finite number of channels with bolometers as detectors are one possibility; there are many papers by the so-called "TICFIRE" project being run out of the Canadian Space Agency that might be useful. But how important is the calibration of cheaper systems like this?

Authors: We have added a few comments indicating additional design factors that could be useful (Section 5.2). More in-depth discussion of instrument development is beyond the scope of this work but is an important subject for future work.

Referee: The authors did show results if the radiance bias was 0.2 RU, but that is a pretty small error – a more realistic error might be 1 or 2 RU. Do these results scale? [For example, even with a carefully calibrated AERI, radiance biases close to 1 RU have been reported – see Delamere et al. JGR 2010. It is hard to imagine that a cheaper radiometer would have better spectral calibration than the AERI.]

[Figure]

Authors: We have added retrievals for radiance biases of 0.5 RU (Supplemental) and 1 RU (main text).

Referee: Some more minor points:

How many streams are being used in DISORT? Fewer streams make the RT code faster, but will decrease the accuracy.

Authors: This is a good point. We generally use 16 streams, but we had previously used fewer for smaller particles (when only a few Legendre moments are needed for the phase function), but after communicating with the DISORT group, we re-ran all cases with 16 streams. This had a small effect on retrievals, and model errors are lower. We have added text to the end of Section 3.3 indicating that DISORT is run with 16 streams.

Referee: Eq 18 is incorrect if gamma > 0. The correct formulation was originally provided by Masiello et al. QJRMS2012, but was also presented in Turner and Löhnert JAMC 2014.

Authors: We use Eq. 18 because iterations are repeated until gamma is negligibly small (<0.01). This is clarified.

Referee: "the ideal range for tau is between 0.4 and 5" (line 425). The authors really should indicate how often this is expected to happen in the Arctic. The Cox et al. JAMC 2014 paper provides some information on this, at least for that site.

Authors: This is a good point. We will add text such as the following to address this:

To get a sense of how common such clouds are, Cox et al 2014 found that at Eureka, Nunavut in 2006-2009, clouds with optical depths of 0.25 to 6 accounted for about 32% of AERI measurements (17% when quality control procedures and a PWV threshold of 1 cm were applied; in this work PWV is as high as 3 cm).

Referee: Is there a minimum number of spectral microwindows that need to be used?

Stated a different way, how do the errors in the retrieved cloud properties change for different microwindow subsets?

Authors: This is an interesting topic. We have tested a second set of wavenumbers and reported differences. We agree that further sensitivity studies along these lines would be interesting; however they are beyond the scope of this work.

---

## Author Comment (AC2) · 20 Aug 2019

Reply to Referee 2. We indicate a referee comment with "Referee" and our response with "Authors".

Referee: The methodology is clearly stated, adequate references are made to work by earlier studies although more recent articles may be available, and the analyses are straight-forward. I am glad to see that the software is being made available to the community as described in Section 7. My view is that the paper is suitable for publication pending relatively minor revisions that address the comments that follow.

If there is one primary suggestion to offer, it would be to compare the cloud properties obtained by this method to those obtained from CALIPSO, where new Version4 products are now available (or to a coincident ground-based lidar if possible). Of particular note is that the V4 products have significant improvements in calibration and cloud/aerosol properties. There will be differences between satellite- and surface-based products that will bear further investigation, but this may be outside the scope of this particular study. CALIPSO cloud products have been used heavily in the development and testing phase of many satellite-based cloud retrieval efforts, especially with the discrimination of cloud thermodynamic phase which is a critical component of the current study. As noted in Section 5.4, imperfect cloud phase discrimination can greatly increase the retrieval errors (lines 500-505).

Authors: This is a good suggestion but is unfortunately beyond this scope of this work. We will compare cloud property retrievals using CLARRA to CALIPSO in future work and thank the referee for this suggestion.

Referee: General comments: Lines 58-62: Two points to suggest here: 1. The authors point out the need for portable, low-cost, autonomous IR spectrometers that can make continuous measurements. But these measurements complement those from polar-orbiting IR spectrometers including IASI (on Metop-A/B/C), AIRS, and CrIS. It would be useful to provide an example where the surface measurements fill the gaps between satellite overpasses. 2. Additionally, a primary benefit to surface-based measurements is that the boundary layer is much better characterized than with profiles inferred from a satellite-based spectrometer. In particular, my impression is that the boundary layer profiles are much improved when temperature inversions are present, and this will impact the cloud properties if the layer is at/below the inversion.

Authors: This is a good point. We will add text such as the following to the introduction:

"Such measurements would be beneficial in a number of ways. They could be used to fill gaps in satellite measurements. For example, cloud properties were retrieved at

[Figure]

Eureka from 2006 to 2009 from AERI measurements made nearly-continuously every ~40 seconds (Cox et al 2014). By contrast, satellite overpasses are typically twice per day. They can also be used to compare to satellite-based measurements. Finally, surface-based instruments are better at characterizing clouds in the boundary layer."

Referee: Section 3: This is a long section (over 6 pages) that discusses the Cloud and Atmo-spheric Radiation Retrieval Algorithm (CLARRA) in quite a bit of detail. Cloud height was discussed in great detail in Rowe et al. (2016) and is not repeated herein. Perhaps the readability would be improved by moving much of the theoretical development into an Appendix.

Authors: We have moved the theoretical development from Section 3.1 to the Appendix (almost 3 pages). To further improve readability, we have reorganized subsections in Section 3 and included the most important information in introductory paragraphs, making clear that the remainder of each section provides additional detail (which can be skipped).

Referee: Minor comments: Line 19: please define exactly what is meant by "mixed phase" - is it a homogeneous mixture of ice and liquid particles or something else?

Authors: Yes. We have clarified this: "Mixed-phase clouds were simulated as an external, homogeneous mixture of liquid and ice particles."

Referee: The word "infrared" appears 26 times in the paper - could contract to IR

Line 47: include more up-to-date L'Ecuyer papers, e.g., "Reassessing the effect of cloud type on Earth's energy balance in the age of active spaceborne observations. Part I: Top-of-atmosphere and surface", by TS L'Ecuyer, Y Hang, AV Matus, Z Wang, in Journal of Climate, 2019. There is also a Part 2 manuscript in review.

Lines 293-294: Radiances are selected in two bands: 400 to 600 cm-1 and from 750to 1300 cm-1. As the method is using selected wavenumbers for each chosen spectral resolution, it would be useful to state them in this paper rather than in the supplemental.

Lines 340; 352; 557: suggest changing "in order to" to "to"

Line 356: change "found such error" to "found such errors"

Authors: Thank you - all of the above changes have been made.

Referee: Lines 503-505: Cloud height: is CO2 slicing used for both water and ice clouds? If so .... might want to change this so it's used primarily for ice clouds and use 11-$\mu$m for optically thick clouds.

Authors: This is an interesting point but is beyond the scope of this work, which focuses more on the microphysical retrievals than the cloud height retrievals (discussed in Rowe et al 2016). We will investigate using 11 micron for cloud height retrievals in future work, and thank the reviewer for this suggestion.

Referee:

Line 536: Polar Regions does not have to be capitalized.

Line 561: suggest changing "correctable" to "mitigated"

Authors: We have made the above changes.

In addition to these changes and changes in response to the other reviewer, we have added a new figure (Fig. 4) and made a number of edits for grammar and clarity. We also made small changes in the retrievals (e.g. standardized number of streams to 16, removed radiance error threshold); resulting differences are minor and do not affect our conclusions.

We thank the reviewer for these helpful suggestions that have improved our paper.